# Modern Trends in Microelectronics Packaging Reliability Testing

**DOI:** 10.3390/mi15030398

**Published:** 2024-03-15

**Authors:** Emmanuel Bender, Joseph B. Bernstein, Duane S. Boning

**Affiliations:** 1Microsystems Technology Laboratories, Massachusetts Institute of Technology, Cambridge, MA 02142, USA; 2Electrical and Electronics Engineering, Ariel University, Ariel 40700, Israel; josephbe@ariel.ac.il

**Keywords:** packaging reliability, solder ball failure, fan-out packaging, silicon interconnect fabric, reliability prediction

## Abstract

In this review, recent trends in microelectronics packaging reliability are summarized. We review the technology from early packaging concepts, including wire bond and BGA, to advanced techniques used in HI schemes such as 3D stacking, interposers, fan-out packaging, and more recently developed silicon interconnect fabric integration. This review includes approaches for both design modification studies and packaged device validation. Methods are explored for compatibility in new complex packaging assemblies. Suggestions are proposed for optimizations of the testing practices to account for the challenges anticipated in upcoming HI packaging schemes.

## 1. Introduction

The ever-increasing demand for more advanced technologies is driving the shrinking of the critical dimensions of integrated circuit (IC) devices. The packaging assemblies of IC technologies must progress, in turn, to accommodate technological progress. Improving package density provides cost-effective performance and capacity improvements. Trends for package miniaturization started in the 1990s, going from wire bonds to solder joints for better performance and I/O pin density. Some steps in their development include PQFP (plastic quad flat pack), SOIC (small-outline integrated circuit), PBGA (plastic ball grid array), and fcCSP (flip-chip–chip-scale packaging) to keep pace with advancements for single chip packages. Performance and throughput were significantly improved in these earlier modified solutions [1,2,3,4].

Lately, solutions for high-density heterogeneous integration (HI), coined 2.5D/3D IC packaging, have been developed. The objective of advanced packaging is IC integration with multiple chips on a package substrate [5]. These orientations often include an intermediate level between the IC and the PCB, called a silicon interposer, to allow the integration of several chips into a single IC structure. This concept has expanded into a large variety of different orientations. Fan-out wafer-level packaging (FOWLP) is one solution. FOWLP is bonded directly onto a circuit board without the need for through-silicon vias (TSVs). A step beyond FOWLP is the fan-out panel-level packaging (FOPLP) solution. FOPLP is an extension of wafer-level fan-out that has a larger substrate size of about 600 mm compared to that of FOWLP, which is about 300 mm. This increases the throughput and lowers the cost-per-device margin compared with FOWLP [6].

The trend of high-density packaging poses new package performance and reliability issues. The reliability of advanced microelectronic packaging has emerged as the top priority across multiple growth markets for semiconductors, including automotive, industrial, and cloud-based computing. The breadth of new device types, including microelectronic logic, memory, power electronics, MEMS, RF silicon-photonics, fluidics, and nanotechnological devices, poses unforeseen challenges to reliability validation engineers [7,8,9]. The reliability of mission-critical packages, some also lifesaving, is of vital importance.

One distinct challenge in performing effective reliability studies on packages is the involvement of an abundance of different materials. Characterization of the assemblies must be preceded by cautious scrutiny of the chemical makeup of the materials and achieving familiarity with the interaction between the different materials. This allows proper planning of what test cases are most effective in causing device failures. Without the preparation needed, testing and simulation results can lead to misleading conclusions. One proposal for significant improvements in packaging reliability testing, initiated by “irel40”, is increased data acquisition in in situ tests [10]. Multiple PCBs are monitored in unison to amass more than 140,000 data points in 1000 temperature cycles. A similar concept can be adopted for the inspection of numerous elements inside advanced packages like micro-bumps or interconnects.

The prime objective of this study is to reveal the most effective testing methods to use for future advanced packaging devices. To achieve this goal, we assess the benefits of testing methods used on conventional packaging devices to perform reliability predictions on the devices, as we previously performed extensively on various electron assemblies [11]. Subsequently, a projection of the uncharted challenges posed by advanced structures is described.

Testing methods are developed to facilitate the exposure of dominant failure mechanisms. In product design, the root causes of failures, based on the physics-of-failure approach, are determined to make an early assessment of the assemblies’ reliability, and improve the design and/or change the materials [12]. This perspective leads to the development of compact models and use of finite element analysis (FEA) studies to reveal the physical composition of the devices. In the final stages of product development, approaches focus on predictive reliability, applied to predict the failure rate or the time to failure in the field, by performing accelerated stress tests. 

Predictive reliability is only effective if it is performed on the foundation of established failure physics. Without sound knowledge of the physics of the devices, the interpretation of the tests results will lead to inaccurate conclusions. For instance, a single stress test will likely accelerate more than one type of failure mechanism. A temperature cycling test will accelerate cracking failures, interfacial delamination failures, as well as fatigue failures. One will not succeed in finding the dominant failure mechanism in used conditions by performing a thermal cycling test on a given device without understanding the underlying physical processes involved. This is because the conditions leading to failure are different for disparate failure mechanisms. It boils down to understanding the root cause of the failures and connecting them to the failure data through a physics-of-failure relation. Figure 1 shows a diagram of the pyramid model for the development of prediction tests based on the underpinnings needed to produce useful results. 

The figure above states the guidelines for a full cycle of testing of a family of device packages. The first stage (bottom level) is to identify the vulnerabilities or failure mechanisms of the materials of which the package is composed. It is especially vital to characterize alterations due to age in the bond interfaces. This information is capsulized into physical models for the individual mechanisms. The next level is the formulation of a model that incorporates interaction of all the failure mechanisms. Linearity is compulsory to allow modularity in the composition of the model. The next stage is the careful selection of test cases based on multiple stress modes to age the packages, apt for activating the different failure mechanisms. The failure data are aggregated into a comprehensive reliability profile for the different stress inputs in the final stage. In HI assemblies, this methodology is essential, and aging data will include the complexity of new phenomena like electrical crosswalk and RF radiation leakage.

There are two overall investigations in this paper. The first is a summary of the reliability concerns and testing methods for standard packaging assemblies. The second is a perspective on advanced packaging reliability issues and proposed strategies on how to characterize their risks. We first lay down some basic models and accelerating factors in Section 2. Section 3 covers many of the reliability studies of conventional packaging assemblies, starting with wire bond packages and accenting the issues with BGA-based packages. Section 4 offers a brief description of methods for packaging mechanism separation. Section 5 elaborates on some reliability studies on advanced packages focusing on future testing possibilities. Our conclusions and final comments are given in Section 6.

## 2. Failure Rate, Time to Failure, and Acceleration Factor Modeling

This section includes a summary of some fundamental reliability prediction terms. The unit of measurement for reliability test results is the failure unit (FIT). FIT is a rate for the number of expected device failures per one-billion-part hours. A system reliability model provides a prediction of the expected mean time between failures (MTBF) for an entire system as the sum of the FIT rates for every component. The failure rate of a single element in a system is defined as
(1)$$FR=\frac{Number\;of\;Fails}{Total\;Unit\;Hours}$$

The failure rate for microelectronic devices can be approximated as the constant time for the period of the product’s useful lifetime. The probability of reliability will follow a first-order Poisson exponential distribution. In the past, ample attention was given to characterizing the decreasing failure rate (DFR) stage of product life. Modern manufacturing processes are very successful at illuminating initial failures by implementing advanced screening and failure mechanism engineering practices in production. The failure rate of the total package or system is expressed as
(2)$$FR_{total}=FR_{temperature}+FR_{humidity}+FR_{voltage}+FR_{cyclic}+\ldots$$

When more than one mechanism exists in a system (almost always the case), the relative acceleration of each mechanism must be characterized under the applied conditions and identified by its unique acceleration factor (AF). FIT values will then be calculated separately for each mechanism at given stress factors. This is possible by isolating different failure mechanisms by applying stress conditions that accent single failure mechanisms. To estimate the FR, the AF needs to be accurately modeled and validated.

Choosing an appropriate value for the AF can be achieved based on the physics of the dominant failure mechanisms occurring in the field. Some of the AF models of the individual stress factors are listed in Table 1.

The essence of failure prediction is the defining of the probability of failure over time. Therefore, failure prediction focuses on the statistical distribution of failures rather than the physical or chemical phenomena that cause the failures. In a bathtub curve, the flat middle section is known as the “useful life” region. There is a relatively constant failure rate (CFR) created by random failures. Over a given period, the occurrence of a failure is unpredictable and independent of prior use.

On the other hand, wear-out failures are often considered life-limiting physical or chemical processes inherently related to the design of the part and its manner of application. Wear-out failures, also called *inherent mechanisms*, generally arise from the interaction of design-related factors and environmental parameters, such as temperature and humidity, and thermal and mechanical cycling loads.

A failure mode is the recognizable electrical symposium by which a failure is observed. A failure mechanism is the specific physical, chemical, metallurgical, environmental phenomena or processes that cause device degradation or malfunction. Failure modes and mechanisms are the end results of the degenerative processes initiated by interactions of the designed and manufactured configuration with the operational and environmental stresses imposed during its period of operation.

Two different stress models are used in calculating the reliability and the time to failure of electronic packages. One is the power law model and the other is the Arrhenius model. Sometimes, these two models are combined to predict the life of components in field application reliability studies. A summary of many common failure prediction models is detailed in Table 2.

## 3. Reliability Analysis and Life Prediction of Packaging Materials and Assemblies

### 3.1. Wire Bond Package Reliability Analysis

Although wire bond packages are mostly replaced with more modern assemblies, many parts still have wire bond packages for some of their ICs. The failure mechanisms observed include bond breakage and delamination. At times, wires break at the heel of wedge bonds due to their reduced cross-sectional area.

In power cycling experiments conducted by Boettge [13], wire bond assemblies failed after approximately 20,000 cycles. The dominant failure mode was bonding wire lift-off. Failure analysis studies revealed residues of bond wire material detected on the surface of the chip metallization. This indicates crack formation and propagation within the aluminum (Al) material of the bonding wire. Cross-sectional analysis of the remaining bond wire contacts also supports this assumption. The crack formation did not occur directly at the interface between the bond wire and chip metallization, but in a shallow layer of the bond wire. Likewise, wires also fractured at the neck of a wire bond. This is different from BGA failures where the resulting fractures are either from tensile or shear forces induced by thermal stress or the flow of the encapsulant during molding.

At times, wire fatigue failures result from interface delamination between the molding compound and the die [14]. One influencing factor of failures seen in Au/Al bonds is the degradation of bonding strength and electrical resistance increase. The increase in resistivity is caused by intermetallic compound (IMC) formation at the Au/Al bonds as well as diffusion at the interface of the Au wire and Ag plating on lead frames. The factors reducing the IMC growth rate are not always found to have a dominant impact on bond reliability; instead, ball bond reliability is dependent on bond parameters and bond pad bendability [15]. However, the presence of the compounds will cause the interface to degrade, often triggering a catastrophic failure.

Khan et al. [16] reported the presence of halogenated organic residues causing increased Au─Al wire bond failures through the degradation of the intermetallic. Results of these failures show an activation energy calculated between 0.7 and 1.0 eV. The E_A_ variation is due to the use of various resins.

Park et al. [17] studied the degradation of the Au─Al bonding under high thermal storage (HTS) reliability testing while using different molding compounds. The lifetime of Au─Al bonding encapsulated by bi-phenyl (BP) epoxy resin is much longer than that of o-cresol novolac (OCN) epoxy resin, and its failure is attributed to the appearance of Sb at the interfaces and bromine (Br) originating from the epoxy mode compounds (EMCs).

Other examples of fatigue failures include wire bond breakage and wire bond lift-off. The wires commonly break at the heel region of the wedge bonds due to a reduced cross-sectional area in that place in the wires. Another failure observed is the disconnection of the heel of the ball bond due to excessive flexing during loop formation and thermal fluctuations during operation.

Fatigue failures due to crack propagation also occur at the wire heel or the neck of the wire bond, leading to an open circuit. These fractures result from either tensile or shear forces induced by thermal stress or the flow of the encapsulant during molding. Occasionally, wire fatigue failures are formed due to interface delamination between the molding compound and the die [14].

One influencing factor of failures seen in Au/Al bonds is the degradation of bonding strength, which causes a resistance increase. This causes IMC formation at the Au/Al bonds as well as diffusion at the interface of the Au wire and Ag plating on lead frames. Regardless of the observations mentioned above, reducing the IMC growth rate does not always have a substantial impact on bond reliability. The dominant factors are the bond parameters and bond strength [15].

To summarize, the key failure mechanisms observed in wire bond assemblies are wire breakage or bond fracture due to stress loads in vulnerable interfaces of the wires. A distinct contrast between wire bond failures and BGA failures, detailed in the next section, is the sensitivity to thermal expansion mismatch of the devices. In BGA assemblies, there are more problems because the BGAs are denser than the wire bond assemblies. Nevertheless, the formation of IMC has proven to have a significant impact on the wire bond interface failures. A wire bond fracture frequently occurs at low temperature cycles, as well.

### 3.2. BGA Package Reliability

#### 3.2.1. Thermal and Thermomechanical Stress Testing on Solder Joints

Flip-chip packaging assemblies include a wide range of vulnerabilities. Figure 2 summarizes common failure modes in flip-chip packages. The effects of these failures are generated using different stress modes, as will be detailed in the following sections.

Zhang et al. shows the consequences of thermal cycling strain on plastic BGA (PBGA) using both an FEA model and Moire fringe pattern analysis techniques in parallel [18]. Bhate et al. presented different techniques for revealing creep and monolithic tests due to shear strain in the solder alloys Sn3.8Ag0.7Cu and Sn1.0Ag0.5Cu using a double-lap shear setup. Their study is reinforced with an FEA inspection of the solder balls with the intention of capturing the change in geometry due to strain [19]. Kumar et al. performed an additional study of lead-free SJ alloys performed on Sn3.8Ag0.7Cu and Sn3.0Ag0.5Cu [20]. Also in this study, the test specimen consisted of four solder joints at the extreme corners of two identical alumina ceramic substrates. The results show that the aging effects are more pronounced in Sn3.8Ag0.7Cu than Sn3.0Ag0.5Cu. Thermomechanical cycling (TMC) was used to accelerate microstructural coarsening in SnAgCu (SAC) solders in [21]. Two coefficients are revealed for the results: a secondary creep derived from a power law and a primary creep from an exponential law.

Since electronic devices consist of a large range of different materials, such as metal, composites, polymers, and ceramics, the thermal expansion of the materials is prone to causing various forms of damage. A prime example is sheer strain in solder joints. Solder joints are subjected to mechanical stresses and strains due to mismatches in the coefficients of thermal expansion (CTE) between the different materials. Components in devices are made up of a variety of materials which have different coefficients of thermal expansion [22]. When a device is switched on and off, it experiences thermal stress, leading to shear stresses, as is demonstrated in Figure 3 between the top and bottom diagrams. Solder joint fractures are categorized according to their modes of failure. Darveaux et al. and Zhao et al. identified three major modes of solder joint failures for (BGAs) [23,24]. The lower solder joint diagram in Figure 3 illustrates the modes as follows:(a)Pad matrix failure. In a study by Henshall et al., pad matrix failures are seen to occur across the matrix layer of the fiber–epoxy polymer composite of the PCB [25]. It is commonly observed as “cratering” on the side of the PCB.(b)Bulk solder failure: Bulk solder failure is the fracturing of the solder sphere. Darveaux et al. showed that this mode is more prone to occuring due to vibration failures [23]. The failure surface tends to be rougher than those seen for the UBM-IMC failures.(c)UBM–IMC failure: Failure cracks usually occur in intermetallic compounds (IMC), which are usually more brittle than bulk solder (Frear et al. 1999), or at the interface with the substrate [26]. They can be identified as a smooth surface on the die substrate or a characteristic ring step and smooth surface at the top of the solder bump.

#### 3.2.2. High-Voltage and -Current-Stressed Solder Joint Studies

Ouyang et al. [27] showed electromigration (EM) wear in 37Pb63Sn flip-chip solder joints which were subjected to constant DC voltage and thermal stress. DUTs were heated in an oven at an ambient temperature of 125 C. The current density at the contact opening due to voltage stress was 1.42 × 10^4^ A/cm^2^.

A notable study showing the effects of electromigration on solder joints from relative current density was presented by P. Dandu et al. [28]. FEA and mathematical models were utilized to quantify the effects of current-crowding and joule-heating the package. The study proposed a modified design where the current enters the bump through a copper trace before being spread through the bulk of the bump. The new design showed a decrease in current density of 42%. In an additional study, an FEA model was used to show the current density singularity in the electromigration of solder bumps [29]. The analyzed structure was simplified to a homogeneous wedge with an arbitrary apex angle: $2\left(\pi -\theta_{0}\right)$. In the results, the current density singularity was observed only at acute angles $\theta_{0}<90^{{}^{\circ}}$.

High-current-stress-induced failures are inspected in µBGAs set to a daisy chain configuration in M. Alam et al. [30]. Due to the low melting point of µBGA, joule heating is an acute concern as well as EM. In [31], H. Gan et al. performed an extensive study on EM in solder joints and solder lines. The study reveals an MTTF of electromigration on two structures: a flip-chip solder/under-bump metallization (UBM) structure and a 3D multilevel aluminum or copper via/wiring structure. The results differ for the structures due to the variation in geometry. Based on the Black equation listed below:(19)$$MTTF=\frac{A}{j^{n}}exp\left(\frac{E_{A}}{k_{B}T}\right)$$
the values 1.8 for n and 0.8 eV/atom for E_A_ are obtained.

#### 3.2.3. Impact and Vibration Stress Studies on Solder Joints

The effects of impact and stress on solder joints can vary significantly depending on the configuration of the PCB. X. J. Fan at el. investigated the influence of the placement of major components corresponding to the placement of secondary components under different drop impact orientations [32]. The results show more severe solder joint failure in off-centered mountings compared to centered assemblies. The consequence of horizonal drops is more acute than that of vertical drops.

The stress distribution on solder joints in aging tests varies with distance from the center of the die. The results from individual solder joints will have a distribution affected by the unequal stress load. This aspect of the results is deceiving when trying to decipher the failure distribution. To generate accurate results, the stress load distribution must be revealed using FEA studies, which compensate for the test results. In [33], an FEA study is performed on SAC SJ pins to find the stress distribution applied to a BGA under vibration stress. The results show the relation between different pins on the edges of the BGA as opposed to those in the center of the BGA.

One notable FEA study on SAC405 solders in a BGA assembly [34] shows that during the solder reflow cooling stage, solder/IMC interface damage was initiated at the most critical solder joint, located underneath the corner of the SI die. However, the interface material point remained intact.

#### 3.2.4. Life-Test Predictions of Solder Joint-Based Configurations

Life-test predictions are the qualification standard for packaged device reliability validation. Table 3 details common life-test prediction types used in industry. These tests, including temperature, pressure, and humidity stress, are administered at accelerated conditions to a large number of packaged devices. In most cases, the tests are expected to be completed without any device failures. There are two points that shed doubt on the ability to accurately qualify the parts based on these tests alone. Firstly, they lack sufficient statistical data. The results are pass/fail. Companies are expected to present zero failures in their results [35,36] to avoid fears that the parts will be unstable in the field. When there are no failures, no statistics are received. Secondly, the tests assume a single acceleration factor. This negates known physics-of-failure models which show that there are multiple failure modes in a device at any given time [37]. Such a method will not locate the dominant failure mechanism in a device. The pass/fail characteristic of the tests is less optimal for generating meaningful data to estimate products lifetimes. Reliability-testing methods usually refer to the JEDEC or Mil-STD-883 standards, as shown in Table 4.

At this point, we will elaborate on the reliability study results of tests using the testing types mentioned in Table 3. A summary of the failure modes corresponding to the accelerated stress tests is presented in Table 5 below. The table also includes descriptions of the effects of the different failures on the devices.

Many details in TMC tests must be understood to allow proper deciphering of the results. Temperature cycling conditions cause complex stress and strain effects on solder joints due to the CTE mismatch between the solder and copper Cu pads/leads of the packaging materials. The consequence is the formulation of micro-cracks on the bond interfaces of the solder joints, leading to a decrease in conductivity and ultimately shorts. The rate of thermal fatigue on solder joints is strongly influenced by the physical parameters of the materials and the BGA configuration and manufacturing.

A study focusing on the formulation of micro-cracks in solder joints was performed by Tu et al. [60]. It revealed that many cracks formed during thermal cycling testing that originated at the intermetallic compound (IMC) η-phase/solder joint interface. The inspection showed that the rate of thickness of the intermetallic layer in the joint directly corresponded to the shortening of the fatigue lifetime of the assemblies. The lifetime of the assemblies was predicted by monitoring the resistance growth inside solder joints during TMC stress. The study concluded that the presence of an IMC causes cracks and affects the fatigue lifetime of solder joints.

Solder joint reliability (SJR) can be improved by using copper posts and increasing the solder tub depth according to Lin et al. [80]. The study concludes that the Cu post diameter is a critical parameter to enhance solder fatigue life. A study by Liu et al. [81] tracks the solder joint fatigue life of ball grid array (BGA) packages under a high cycle vibration load. The results show a consistent trend of crack development with a primary crack originating at the inner corner of the component side, and secondary cracking at the outer corner of the joint. The delamination between the solder mask and solder joint shows clear evidence of the primary crack. No correlation is observed between the crack growth rate compared to the frequency variations of the test.

Noctor et al. [82] evaluated solder attachment reliability for thin small-outline packages (TSOP) under temperature cycling. Delamination of the TSOP sides was observed as well as individual solder joint cracks. The thermal expansion mismatch between the package and the PCBs was the chief cause for these failures. For any particular components, the predicted failure probability depended on the assembly parameters (solder joint dimensions, solder alloy, board CTE, size, and thickness), the intended field environment, and the intended product design life. Accelerated thermal cycling (ATC) tests and predicative modeling showed that TSOP solder joint life during testing was about five times longer with copper Cu than with alloy-42 lead frames.

Jeon et al. [83] studied the relationship between SJR and the properties of a pad surface finish, such as ENIG and OSP. The IMC thickness could be controlled by the thickness of an electroless Cu layer. Amagai [84] demonstrated that a higher CTE and elastic modulus die-attach film can increase the life of solder joints.

With more components classified for portable applications exposed to shock and vibration environments, dynamic loads (often high-cycle-fatigue loads) have significant effects on solder joint fatigue life. The failure mode is observed as cracks on the solder joints as well. Wang et al. [62] studied the vibration fatigue failures seen on solder joints. Fatigue failure was related to the devices’ location in the PCB as well as the bump location on the package. Often, cracks induced by vibration fatigue were created in metal compound layers or solder materials nearby. It was also observed that cracks were first observed in the bottom round-angle area of the joint, and then, appeared at the top of the joint throughout the progress of the vibration. The solder joint was in a failure condition due to the full cracks at the top of the solder joint. The solder joints’ fatigue was also connected with the mass of the chips, the stiffness of the chips, and the shape and number of solder joints.

#### 3.2.5. Device Failures in the Field

Gauging the success of reliability testing efforts can take place by assessing field experience. Figure 4 depicts a relative study on field failures. This analysis will contribute to future testing programs to adjust testing to mitigate failures of this kind.

#### 3.2.6. Failure Mechanism Coverage of Different Aging Tests

Failure mechanism observation will be different depending on the selection of the test conditions. Figure 4 shows the breakdown of failure mechanism observation found in packaged devices exposed to two different tests. The first group was subjected to temperature cycling with a thermal swing of −65 °C to 150 °C. The dominant failure mechanism observed was bond failure, followed by die cracks. The thermal expansion offset of the die compared to the PCB, as demonstrated in Figure 3 above, is the source bond failures and die cracks. The second set of devices were subjected to HAST stress conditions of a 130 °C ambient temperature and 85% relative humidity (RH). As expected, the great majority of failures observed were due to moisture penetration [85].

It is also true that different testing types can accelerate the same failure mechanisms. With that in mind, one must take care to make scrupulous planning regarding test suites to choose relevant test cases to magnify all major failure mechanisms. The level of stress is also crucial. For instance, HAST tests under conditions of 130 °C/85%RH caused many failures, as is demonstrated in Figure 5, but HAST tests with conditions of 85 °C/85%RH resulted in almost no damage to the samples [86].

#### 3.2.7. Aging and Prognostics Testing of Solder Joints

Valuable knowledge about packaging longevity is acquired from the aging and prognostics monitoring of manufactured devices. The disadvantage of validation tests with pass/fail criteria, mentioned in the previous section, include the lack of pre-emptive failure information. A solution for this issue is testing processes that monitor the characteristics of failure modes preceding the failures. Gershman et al. studied crack propagation quad flat no 44 lead (QFN44) packs using the nondestructive quantitative analysis of crack propagation in solder joints (NDQC) technique [87]. Their study shows that the gradual progression of cracks on the packages can be characterized early in their formation. In [88], a crack propagation study was conducted with an analytical model and an FEA study in early failure prediction. An accurate fit was received between the analytical model and the FEA model. The results of the two previous studies are formulated into a health-monitoring technique in [89].

An additional solution for early failure discovery is proposed by Hofmeister et al., coined Ball Grid Array (BGA) Solder Joint Intermittency Detection: SJ BIST™ [90]. This method is intended for both the fault coverage and health management of solder joints in FPGA boards. SJ BIST detects high-resistance faults in the solder joint networks of operational FPGAs.

## 4. Methods for Separation of Failure Mechanisms in Packaging Assemblies

This section details methods dedicated to revealing and profiling multiple failure mechanisms reacting simultaneously in packaging assemblies. Failure data commonly contain a mixture of different failure mechanisms. This is the product of several different failure mechanisms being accelerated by the same stress conditions. For example, stress voiding and creep will result in excess expansion and contraction of the materials, which are accelerated with temperature cycling. The characteristics of the different FMs can only be realized after separating them from resulting failure distributions. The distributions will have discrepancies due to the mixing of the mechanisms. The distributions are deconvolved to reveal the distributions for the separate failure mechanisms using competing risk models, as detailed in the literature [91,92]. U. Chakraborty et al. [93] used differential evolution (DE) and the L-BFGS-B (limited-memory Broyden–Fletcher–Goldfarb–Shanno with the box constraint) method to separate mechanisms based on completing risk models on simulated electromigration and stress voiding data. The results confirm that this approach is highly precise in parameter identification from lifetime data at different temperatures.

## 5. Latest Reliability Testing Studies on Advanced Package Assemblies

### 5.1. Outstanding Reliability Concerns in Advanced Packages

Reliability prediction in advanced packages differs from that in conventional packages both in terms of complexity and with regard to the combination of a wide spectrum of diverse elements. For this reason, modern package development is referred to as heterogeneous integration (HI). Hybrid bonds in advanced packages are created using new processes to allow highly dense interconnect fabrication. The consequences for reliability are numerous [94]. One major risk in new packages is their increased dimensions and stacked layers. This results in increased delamination and warping problems [95]. One thread of HI is aimed at the development of system-in-package (SiP) devices. SiP technology has been accompanied by many reliability issues. Due to the augmentation of density and power, thermal management issues become a crisis [8]. The number of interconnects involved in the new packages can be higher than in previous packages by two orders of magnitude [96]. Along with concerns of bonding failures, wafer-to-wafer alignment challenges in 3D integration have yet to be fully resolved [97]. This problem is magnified due to thermal misalignment with time [98,99,100,101].

The reliability risk of packages due to misalignment must be carefully evaluated. In Chip-on-Wafer-on-Substrate (CoWoS) and other technologies, micro-bumps (µBumps) are used for interconnecting the chiplets. The Silicon Interconnect Fabric (Si-IF) process is used to fabricate the chiplets with µBumps. In this process, heterogeneous components (chips, passives, etc.) are mounted on Si-IF at 2–10 µ interconnect pitch constraints (very dense) [102]. The control of µBump coplanarity is a reliability risk due to increased chip size, non-uniform µBump layout, and higher µBump density [103]. Figure 6 displays a model of a CoWoS architecture.

A distinct family of reliability issues resides in the implementation of thin wafers in advanced packages [104]. Three-dimensional IC integration is enabled due to wafer thinning. Wafer thicknesses are approaching 10µ [105]. This results in additional mechanical stress and strain to the wafer. Some of the failure risks include wafer warpage, interconnect/Si cracking and delamination, and deterioration of the device characteristics.

As electronic systems in automobiles and airplanes, etc., become more complex, the use of advanced packages is more rampant. As a result, their technologies must be capable of enduring harsh environments. With autonomous control, the operation hours can be significantly increased [106].

One of the distinct differences in material compounds in advanced packages is the transition to low-temperature solders (LTS) such as Sn–Bi and In-based solders. LTSs, having a lower melting temperature than their SAC counterparts, mitigate warpage during the board-attach reflow process [107]. Another concern in high reflow temperatures is delamination in hybrid bonds between different tiers [108]. The characteristics of LTS materials have reliability challenges like EM and thermomechanical failure [109,110].

**Figure 6 micromachines-15-00398-f006:**
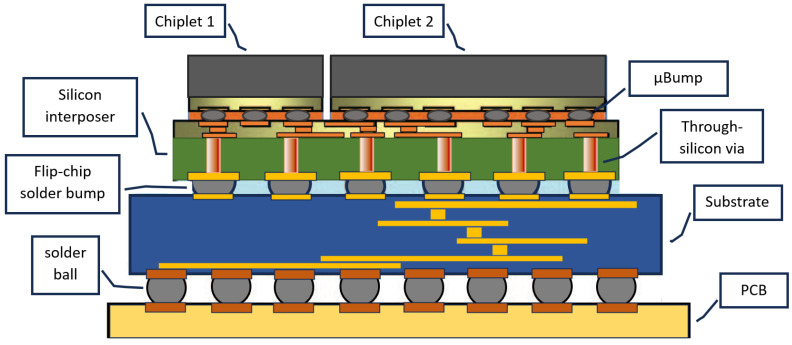
Diagram of a CoWoS package. Inspired by [109].

### 5.2. Reliability Analysis and Testing Performed on Advanced Packages

Many reliability studies have been performed on advanced package assemblies. Table 6 gives some examples of recent tests and simulations. The testing practices show strong correlations with the procedures used in standard packages. The shortcomings of the standard testing practices today were mentioned earlier in this work. The lack of statistical detail in the results (being pass/fail criteria) are a cause of uncertainty of the tests’ ability to locate real reliability threats in packaging devices. In standard packages, qualification tests are more trusted because of the amount of test data assumed over the years and the relative simplicity of the packages. In advanced packages, both points are lacking.

Successful reliability methods for advanced packaging qualification should obtain a vivid picture of all the hazards in the system corresponding to their relative impacts. This is only possible by obtaining failure data of multiple elements in multiple stress modes. Multiple elements can be obtained by probing multiple signal lines in parallel and monitoring the degradation over time. Statistical distributions of the degradation/failure data can be used to determine early failures and mean failure times. A significant challenge in the process is the separation of multiple mechanisms in the data. Without mechanism separation, failure rates and times are impossible to decipher.

In parallel to this study, the authors are proceeding to validate a method for monitoring multiple pins in a BGA package. The pins will be probed to determine the degradation over time and failure. The results will be analyzed with mechanism separation methods to find the relative impact of the different mechanisms. Similarly, this methodology can be adopted and incorporated in advanced assemblies. More information about these methods will be provided in future studies.

## 6. Conclusions

We have summarized many common microelectronic packaging reliability testing trends. The testing methods assessed in this work originated in older packaging assemblies and have been adopted in more recent orientations. The objective of this study is to create a perspective of testing practices that have a high probability of success in advanced packaging schemes. This is with the understanding that advanced packages are far more complex and include a broader range of hybrid bonds. Our suggestions are as follows:
(1)Failure mechanisms and testing methods of standard packaging schemes are considered acceptable qualification metrics for advanced packages based on most current studies.(2)Accurate reliability prediction in advanced packages will require testing methods more optimized than the standard methods used for conventional assemblies. The current testing standards propose product qualification based on accelerated tests in set stress conditions. The products pass the test if they endure the stress conditions after a given time.(3)Testing methods with single stress modes assume a single failure mechanism in the device. This assumption is observed to be inaccurate. Successful testing methods must succeed in revealing and separating multiple failure mechanisms.(4)Methods are in development for extraction and analysis of multiple data elements of standard and advanced packages. By deciphering the failure characteristics of the parts from the statistics of the results, accurate failure rates and failure times can be achieved. This will raise the bar for the reliability testing of advanced assemblies and complete systems.

## Figures and Tables

**Figure 1 micromachines-15-00398-f001:**
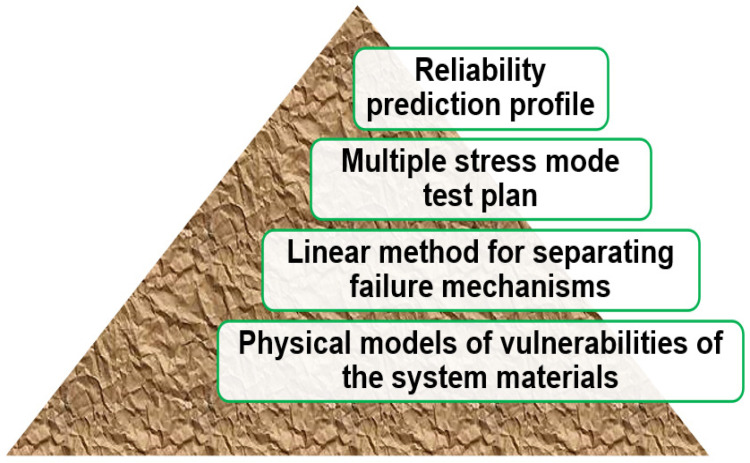
“Building from ground up”—pyramid approach.

**Figure 2 micromachines-15-00398-f002:**
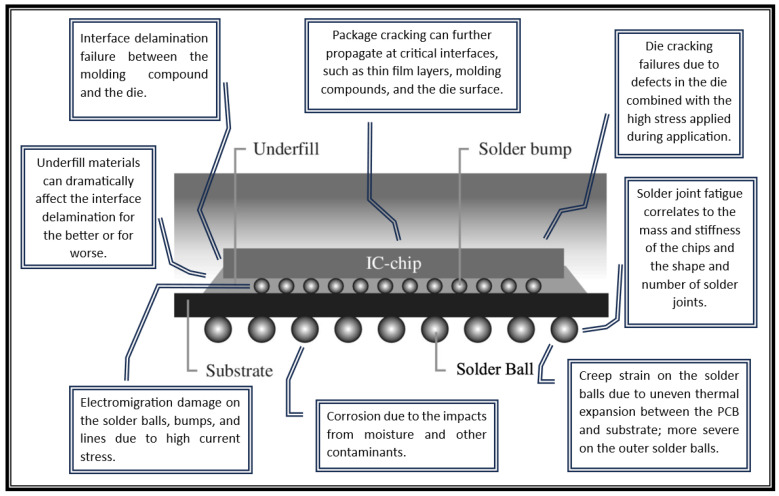
Different failure modes in flip-chip devices.

**Figure 3 micromachines-15-00398-f003:**
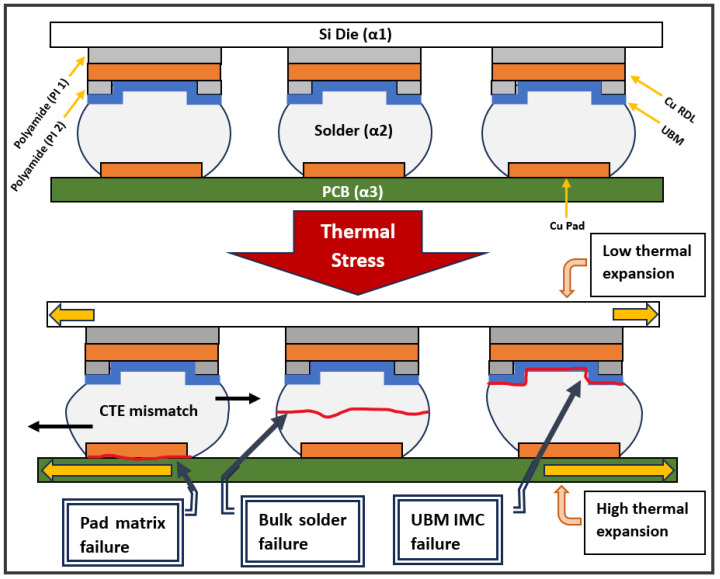
Solder interconnects subjected to shear stresses during thermal cycling due to CTE mismatch between the die (α1), solder (α2), and substrate (α3). α denotes the CTE of the material.

**Figure 4 micromachines-15-00398-f004:**
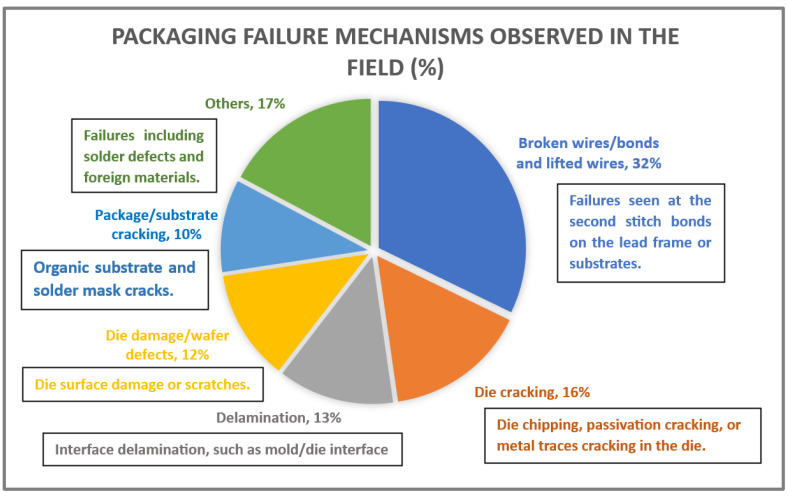
Pie chart showing the breakdown of failures observed in the field categorized by failure mechanism [11].

**Figure 5 micromachines-15-00398-f005:**
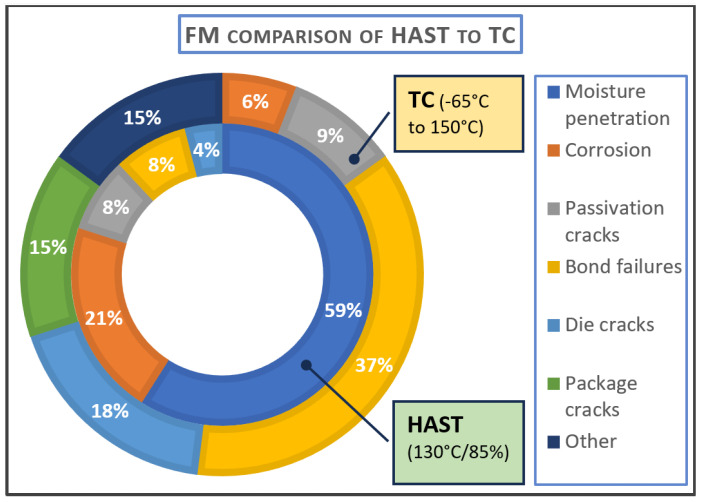
Doughnut chart comparison of failure mechanisms detected by HAST and TC [11].

**Table 1 micromachines-15-00398-t001:** Common models for failure mechanism acceleration.

Model	Acceleration Factor	Parameters
Temperature (Arrhenius model)	$AF(T)=\exp\left[\frac{E_{a}}{k}•\left(\frac{1}{T_{use}}-\frac{1}{T_{stress}}\right)\right]$ (3)	*E_a_*: activation energy *k*: Boltzmann constant = 8.617 × 10^−5^ eV/k *T*: temperature, K
Temperature and voltage (Eyring model)	$AF(T,V)=AF(T)•\exp\left(B•\left(V_{stress}-V_{use}\right)\right)$ (4)	*B* depends on mechanism, default *B* = 1. *V*: voltage, V
Temperature and relative humidity (model for corrosion failures in plastic packages: Peck model)	$AF(T,RH)=AF(T)•{\left(\frac{RH_{stress}}{RH_{use}}\right)}^{n}$ (5)	*n* = 3, *E_a_* = 0.9 eV *RH*: relative humidity, %
Temperature cycling (model for mechanical fatigue failures of solder/other contacts: Coffin–Manson model)	$AF(\Delta T)={\left(\frac{\Delta T_{stress}}{\Delta T_{use}}\right)}^{C}$ (6)	*C* depends on material’s mechanical properties. ΔT: temperature interval, K or C.
**Vibration, drop, and strain testing (model for accelerated** shock and strain on solder joint, interconnects, etc.)	$AF\left(\Delta g\right)=\frac{g_{rms\;t}}{g_{rms\;f}}=\sqrt{\left(\frac{t_{f}}{t_{t}}\right)}$ (7)	*g_rmst_* = stress test conditions [m/s^2^] *g_rmsf_* = stress use conditions [m/s^2^] *t_f_* = use condition stress time t_t_ = test condition stress time

**Table 2 micromachines-15-00398-t002:** Common failure prediction models used in packaging testing.

Model Name	Description/Parameters	Application Examples	Model Equation
Coffin–Manson	Failure time estimation in thermal cycles	Fatigue of solder joint and other connections	$N_{f}=C{\left(\frac{\Delta \gamma }{2\varepsilon_{f}}\right)}^{1/B}$ (8) where *N_f_* _=_ mean number of cycles to failure Δγ = inelastic strain range, where $\Delta \gamma =F\frac{L_{D}}{h}\Delta \alpha \Delta T$ $\varepsilon_{f}$ = fatigue ductile coefficient in shear, e.g., solder constant is 0.325 B = scale factor determined by experiment C = fatigue strength coefficient
Norris and Landzberg	Life as a function of thermal cycles	Thermal fatigue of tin-lead solder interconnects	$\frac{N_{lab}}{N_{machine}}={\left(\frac{f_{L}}{f_{M}}\right)}^{1/3}{\left(\frac{\Delta T_{M}}{\Delta T_{L}}\right)}^{2}\Phi \left(T_{max}\right)$ (9) where $\frac{f_{L}}{f_{M}}$$\;\mathrm{and}\;\frac{\Delta T_{M}}{\Delta T_{L}}$ are the maximum fatigue life and temperature change ratios under isothermal conditions, $\Phi \left(T_{max}\right)=\exp\left(\frac{E_{a}}{k}\left(\frac{1}{T_{Lmax}}-\frac{1}{T_{Mmax}}\right)\right)$
Miner’s Rule	Cumulative linear fatigue damage as a function of flexing	Metal fatigue (valid only up to the yield strength of the material)	$CD=\sum_{i=1}^{k}\frac{C_{Si}}{N_{i}}\leq 1$ (10) where CD = cumulative damage Cs = number of cycles applied @ stress Si Ni = number of cycles to failure under stress Si (determined from an S-N diagram for that specific material) k = number of loads applied
Coffin–Manson	Fatigue life of metals (ductile materials) due to thermal cycling and/or thermal shock	Solder joints and other connections	$Life=\frac{A}{{\left(\Delta T\right)}^{B}}$ (11) where Life = cycles to failure A = scale factor determined by experiment B = scale factor determined by experiment ∆T = temperature change
Plastic strain	Plastic strain-based life prediction model based on the power law.	Typically used in bulk solder failure	$D_{f}=A{\left(\varepsilon_{p}\right)}^{b}$ (12) where D_f_ = number of drops to failure A is a constant b is an exponent ε_p_ = plastic strain
Garofalo	Very slow vibration which initiates creep strain	Lead-free electronic interconnects	${\dot{\varepsilon }}_{cr}=C_{1}{\left[sinh\left(C_{2}\sigma_{e}\right)\right]}^{C_{3}}exp\left(C_{4}/T\right)$ (13) where $\sigma_{e}=$ creep strain constant C1–C4 = creep parameters dependent on the implicit creep model B = scale factor determined by experiment T = temperature
Peck’s	Life as a combined function of temperature and humidity	Epoxy packaging	$\tau =A_{0}{\left(RH\right)}^{-2.7}\exp\left[\frac{0.79}{kT}\right]$ (14) where t = median life (time to failure) A_0_ = scale factor determined by experiment RH = relative humidity k = Boltzmann’s constant = 8.62 × 10^−5^ eV/K T = temperature (degrees Kelvin)
Peck’s Power Law	Time to failure as a function of relative humidity voltage and temperature	Corrosion	$TF=A_{0}\cdot RH^{-N}\cdot f\left(V\right)\cdot \exp\left[E_{a}/kT\right]$ (15) where TF = time to failure A_0_ = scale factor determined by experiment RH = relative humidity N = ~2.7 Ea = 0.7–0.8 eV (appropriate for aluminum corrosion when chlorides are present) f(V) = an unknown function of applied voltage k = Boltzmann’s constant = 8.62 × 10^−5^ eV/K T = temperature (degrees Kelvin)
Eyring/Black/Kenney	Life as a function of temperature and voltage (or current density (Black))	Capacitors, electromigration in aluminum conductors	$\tau =\frac{A}{T}\exp\left[\frac{B}{kT}\right]$ (16) where t = median life (time to failure) A = scale factor determined by experiment B = scale factor determined by experiment k = Boltzmann’s constant = 8.62 × 10^−5^ eV/K T = temperature (degrees Kelvin)
Eyring	Time to failure as a function of current, electric field, and temperature	Surface inversion, mechanical stress	$TF=B{\left(I_{sub}\right)}^{-N}\exp\left(E_{a}/kT\right)$ (17) where TF = time to failure B = scale factor determined by experiment I_sub_ = peak substrate current during stress N = 2 to 4 Ea = −0.1 to −0.2 eV (note that the apparent activation energy can be negative) k = Boltzmann’s constant = 8.62 × 10^−5^ eV/K T = temperature (degrees Kelvin)
Thermomechanical Stress	Time to failure as a function of change in temperature	Stress generated by differing thermal expansion rates	$TF=B_{0}{\left(T_{0}-T\right)}^{-n}\exp\left(E_{a}/kT\right)$ (18) where TF = time to failure B0 = scale factor determined by experiment T0 = stress free temperature for metal (approximate metal deposition temperature for aluminum) N = 2–3 Ea = 0.5–0.6 eV for grain-boundary diffusion, ~1 eV for intra-grain diffusion k = Boltzmann’s constant = 8.62 × 10^−5^ eV/K T = temperature (degrees Kelvin)

**Table 3 micromachines-15-00398-t003:** Examples of standard life-test qualifications.

Test Types	Stress Conditions	Test Duration/Accept	Sample Size	Results
Preconditioning test	30 °C/60%RH	200 h	Sum of samples for TC and HAST	Pass/fail
Temperature cycling	−65 °C to 150 °C	1000 cycles	45 units per lot for 3 lots	Pass/fail
Temperature and humidity test (no bias)	85 °C/85%RH	1000 h	45 units per lot for 3 lots	Pass/fail
Pressure cooker test (PCT)	121 °C/2atm/100%RH	200 h	45 units per lot for 3 lots	Pass/fail
Highly accelerated stress test (HAST)	130 °C/85%RH	100 h	45 units per lot for 3 lots	Pass/fail
Thermal shock	−55 °C to 125 °C	1000 cycles	45 units per lot for 3 lots	Pass/fail
High-temperature storage	150 °C	1000 h	45 units per lot for 3 lots	Pass/fail
Solder Ball Shear (SBS)	Mechanical shear stress	1000 h	30 bonds/5 units	Pass/fail

**Table 4 micromachines-15-00398-t004:** Standard test practices [38].

Test	Test Conditions in JEDEC	Target Failure Mechanism
Preconditioning	JESD22A 113	Cracking, delamination, interconnect damage failures
Unbiased and biased highly accelerated stress testing (HAST)	JESD22A118	Corrosion, delamination, contamination, and migration; polymer aging failures
High-temperature storage	JESD22A103	Diffusion, oxidation, degradation of material properties, IMC, creep failures
Temperature humidity bias (or no bias) (THB)	JESD22A101	Corrosion, contamination, and migration failures
Temperature cycling (TC)	JESD22A104	Cracking, deamination, fatigue failures
Power thermal cycling	JESDA105	Cracking and delamination, fatigue, material degradation failures
Mechanical shock (drop)	JESD22B104	Cracking and delamination and fatigue, brittle fracture failures
Vibration	JESD22-B103B	Solder joint failures, and cracking and impact failures
Bending	JESD22B113	Package, solder joint failures, cracking, and delamination
Thermal shock (TS)	JESD22A106	Cracking, delamination, and fatigue; brittle fracture failures
Autoclave (PCT)	JESD22A102	Corrosion, delamination, and migration; interface contamination failures
Solder Ball Shear (SBS)	JESD22B117	Solder joint failures and I/O shorts
Solderability (SD)	JESD22B102	Solder joint failures and creep failures

**Table 5 micromachines-15-00398-t005:** Studies to reveal failures using accelerated tests.

Packaging Failure Mechanisms	Failure Mechanism Descriptions	Accelerated Stressors	Sources
Die cracking; thin film cracking; passivation cracking	Serious decrease in performance and, at times, open-circuit failures	Temperature cycling; power cycling; thermal shock and preconditioning test. Example conditions are −55 °C +125 °C and 65 °C +150 °C	Merrett et al., 1983 [39], Shirley et al., 1987 [40], Blish et al., 1991 [41], Hu et al., 1995 [42], Annaniah et al., 2017 [43], H Zhou et al. 2023 [9]
Interface delamination and induced micro-cracks	Delamination and cracking inside the die or any other interfaces in the package	Temperature cycling and thermal shock; HAST; temperature and humidity test; pressure cooker test; and mechanical bending test in stacked-die chip-scale packages (CSPs)	Emerson et al., 1994 [44], Tanaka et al., 1999 [45], Aihara et al., 2001 [46], Chung et al., 2002 [47], Saitoh et al., 2003 [48], Kwon et al., 2005 [49], Braun et al., 2006 [50], C. Qin et al., 2020 [7]
Bond pad crack	Gap between the epoxy and die top, detecting wire bond inter-layer dielectric crack using dark-field imaging	Low-k device bond pad, crack post temperature cycle; root cause for Al bond pad crack post TC; local compressive/tensile loading during wire bonding impact/vibration step	Liu et al., 2019 [51], Boettge et al., 2018 [13], Kho et al., 2021 [52], H. Zhou et al., 2023 [9]
Package cracking; substrate cracking; underfill cracking	Package body or internal “element” cracking	Temperature cycling, such as −65 °C +150 °C; impact of package geometry on delamination	Zelenka et al., 1991 [53], Amagai et al., 1995 [54], Dias et al., 1997 [55], Ahn et al., 2000 [56], Lin et al., 2005 [57], Mercado et al., 2003 [58]
Solder joint fatigue/cracking; BGA and PoP ball failure	Solder joint cracking and solder creep fatigue damage	Temperature cycling; power cycling; vibration fatigue testing	Tu et al., 1997 [59], Suhling et al., 2004 [60], Wang et al., 2004 [61], Birzer et al., 2006 [62], Davis et al., 2007 [63], Ghaffarian et al., 2019 [64]
Wire lifting/broken bond/heel broken of stitch bonds	IMC cracks or wire heel cracking and bond degradation	High-temperature storage (150 °C, 170 °C); power cycling and thermal cycling	Uebbing et al., 1981 [14], Hund et al., 1985 [15], Wu et al., 1995 [65], Cory et al., 2000 [66], Park et al., 2004 [17], Tang et al., 2020 [67], Xu et al., 2021 [68]
Corrosion	Due to the impacts from moisture and contaminants; due to the residues present on the electronic device (PCBs)	Temperature and humidity test; pressure cooker test; HAST; PCT	Striny et al., 1981 [69], Emerson et al., 1992 [70], Pecht et al., 1995 [71], Tran et al., 2000 [72], Wagner et al., 2014 [73], C. Qin et al., 2020 [7],
Electromigration	Damage seen at interconnects or solder bumps with high-current applications	Current density; temperature; directionality of EM failure at high current density and temperature conditions	Wu et al., 2004 [74], Balkan et al., 2004 [75], Shao et al., 2004 [76], Basaran et al., 2005 [77], Ding et al., 2005 [78], Tajedini et al., 2021 [79]

**Table 6 micromachines-15-00398-t006:** Recent reliability studies on advanced packaging assemblies.

Topic	Device Tested	Test Type	References
Thermomechanical reliability of an aWLP fan-out package	aWLP (advanced wafer-level package)	FEA modeling of the creep strain energy density (CSED) to die and package dimensions	[111]
2.5D packaging development	CoWoS MCM	FESA to check warpage and thermal cycling	[101,112]
CoWoS architecture in 2.5D system	CoWoS-S compared to CoWoS-L	μ-bump, TSV, TIV, and C4 daisy chains	[113]
3D die-to-wafer hybrid Cu bonding	HCB with 4 μm pitch and 2 μm pad	Daisy chains that consist of 2 μm pad in 4 μm pitch	[114]
Wafer-level chip-size package (WLCSP) reliability	WLCASP packages designed with PBO2 openings: 130 µm and 190 µm	Thermal shock test (LLTS)—LLTS75x stress	[115]
Reliability of fan-out WLP	FOWLP packages	Standard JEDEC reliability tests	[6]
Electroplating uniformity in FOPLP	FOPLP packages	FEA simulations of thickness variation	[116]
Electromigration reliability of Cu redistribution line (RDL) technology	20 μm long Cu RDLs	Failure analysis of high-current and-temperature-stressed devices and FEA models	[117]
Parylene-HT as dielectric compared to SiO_2_	Mirco-vias surrounded by dielectrics	FA models to simulate thermomechanical strain	[118]
Reliability of high-layer-count PCBs	Large-area fan-out package	Standard JEDEC reliability tests	[119]
Silicon interposer fabrication and reliability	Large silicon interposer	Thermal cycling	[102]

## Data Availability

Not applicable.

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
