# Peer review of "Modern Trends in Microelectronics Packaging Reliability Testing"

_micromachines, 2024, doi:10.3390/mi15030398_

Round 1

Reviewer 1 Report

Comments and Suggestions for Authors

The authors reviewed the reliability in microelectronics packaging and summarized the technologies for the packaging reliability characterization, as well as the trends of the technology evolution towards advanced packaging. They categorized the failure mechanisms, test methods, as well as many examples covering the past, today, and tomorrow’s packaging. I believe it is useful for readers from both academic and industry.  One suggestion is that the contents for the advanced packaging can be increased and more details and typical examples would be interesting for readers, since this is today’s mainstream of packaging. In the current version this section is not in detail, although there is a summary including abstracts of some references. 

Comments on the Quality of English Language

The authors should carefully check the whole manuscript to correct any errors. Just list some typos (not include all):

1. page 2, line 71;

2. page 7, line 154;

3. page 13, line 318, the abbreviation “BGA” should be explained when it appeared the first time, and line 334;

4. again, page 15, line 389;

5. page 16, line 418.

Author Response

Dear Reviewer,

Please excuse the mistake in the file the was uploaded to you. The file was meant for a different reviewer. Please see the modified manuscript.

Sincerely,

The Authors

Reviewer 2 Report

Comments and Suggestions for Authors

The authors presented a comprehensive review of standards, methods, and notable research works on testing, and packaging reliability of microelectronic devices. The paper is well written.

Though the authors provided a good review of existing and emerging packaging reliability and test methods to improve them, the review deems incomplete as it does not adequately address the packaging reliability and testing challenges of current and future technology trends.

The paper can be accepted if the authors address the following concerns:

1.     In Section 2, Tables 1 and 2 are focused mainly on thermal stress related failure mechanism of wire bonds, solder balls, etc. However, in current densely populated packages, thermal stress is not the only issue of package reliability. Electrical crosstalk, dielectric charging, leakage current are major sources of package failure and reliability degradation in close proximity nanometer scale technology nodes. Vertical interconnects and RDL levels in high-density high-speed multi-level 3D packaging contribute to the mentioned electrical reliability challenges, in addition to the common thermal stress related mechanical reliability issues. If the package includes high frequency RF modules (RFICs), high frequency leakage through vias aggravate the reliability problem further. The authors need to include current research on these reliability issues.

2.     The authors mentioned in several places abut solder ball failures, micro-cracks, die cracks, and delamination. However, there is not a single SEM or conceptual diagram illustrating such scenarios. The authors must include either SEMs or conceptual diagrams of major failure modes.

3.     The authors touched on FEA studies to predict the failure modes and mechanisms. However, no FEA simulation results were provided. A comparison, including the challenges, of the capabilities of various types of FEA software in modeling different types of failure mechanisms for microelectronic devices as the technology nodes are shrinking can be included as a table.

4.     The failure modes of emerging wafer level packaging technology are different than conventional wire bonding or solder ball bonding. The authors need to include this in their analysis to better capture the trends.

5.     In heterogeneous integration of diverse technology dies, the failure mechanisms of different types of dies are different. For example, a certain level of stress may be acceptable for a specific technology (or technology node) but may be devastating for another type of device (or technology node). If an SiP includes microelectronic logic, memory, power electronics, MEMS, RF, photonics, fluidics, and nanotechnological devices, and all are necessary to be packaged in wafer level, the failure modes and mechanisms of each type of devices must be incorporated in the reliability model and life time estimation. This review is missing.

6.     It is better to include how much of the current testing Standards (table IV) applies to heterogeneous integration packages including wafer level packaging, and what can be done to bridge to gap.

7.     One of the current trends is to reduce the interconnect size by a factor of 20 from BGA to microbumps. However, the review does not include any discussion on microbumps.

8.     It is better to include heterogeneous integration in the pyramid model. As the pyramid model is a one–dimensional approach, for heterogeneous integration, some sort of parallelism is necessary to be incorporated in the model to capture the heterogeneous integration nature of the package.

9.     Electro-mechanical failure of vias is a crucial issue in 3D packaging and needs to be included in the review.

10. Table IV provides a good summary of testing standards; however, the listed failure mechanisms are all of mechanical origin. What are the standards for dielectric charging, leakage current, crosstalk, RF radiation leak, electrostatic and magnetic field induced reliability degradation?  

11. Most of the references in Table V are pretty old except a few.

Author Response

Dear Reviewer,

The Authors had a pleasure to see that you took the time to read and understand our paper so deeply. The comments and suggestions made are vary valuable. The authors make a concentrated effort to address all of your comments and suggestions. The authors hope that the changes in the manuscript will be well accepted.

Sincerely,

The authors of the manuscript. 

Reviewer 3 Report

Comments and Suggestions for Authors

My comments are annotated into the PDF.

The paper is far from being complete, and the title is misleading. So I suggest the authors give some more focus on the story.

Author Response

Dear Reviewer,

It was a pleasure to receive your comments and suggestions. Please find the modified PDF file that you sent with our responses.

Sincerely yours,

The authors 

Round 2

Reviewer 2 Report

Comments and Suggestions for Authors

The authors made some changes to improve the paper. The reference section has been improved considerably. Though the authors did improve the paper, they left the reliability issues of advanced state-of-the packaging as part of future research, or touched very lightly. That makes the paper an overview (not review) of current packaging practices, not Modern trends, with very little advancement, like an undergraduate textbook. Nonetheless, some may find the paper useful in limited purposes..

Reviewer 3 Report

Comments and Suggestions for Authors

One last remark: the tittle is incorrect. The word testing should be added.

Comments on the Quality of English Language

One last remark: the tittle is incorrect. The word testing should be added.